# Changes in intraocular pressure before, during, and after playing Thai Traditional, Thai Folk, and Western wind instruments

Jirach Jatechayanon[1], Somkiat Asawaphureekorn[1], Piyanan Suparattanagool[2], Sukhumal Thanapaisal [1,3]*

1 Department of Ophthalmology, Faculty of Medicine, Khon Kaen University, Khon Kaen, Thailand, 2 Clinical Epidemiology Unit, Faculty of Medicine, Khon Kaen University, Khon Kaen, Thailand, 3 KKU Glaucoma Center of Excellence, Department of Ophthalmology, Faculty of Medicine, Khon Kaen University, Khon Kaen, Thailand

* sukhth@kku.ac.th

## Abstract

### Objectives

To compare and evaluate intraocular pressure (IOP) during and after playing Thai traditional (TT), Thai folk (TF), and Western (WT) wind instruments.

### Methods

Three types of wind instruments were used: Thai traditional, Thai folk, and Western, played by three groups of players according to an identical set of notes for 120 seconds. Five IOP measurements were performed before, during (37 and 101 seconds), and after (150 and 240 seconds) playing the instruments. Primary outcomes were differences in IOP between groups during the performance. Secondary outcomes were differences in IOP between groups after the performance. A mixed-effect model for repeated measures (MMRM) was used for statistical analysis.

### Results

Thirty eyes from 15 participants were included, with mean (SD) baseline IOP of 13.3 (2.7), 12.3 (1.5), and 14.5 (2.4) mmHg in WT, TT, and TF groups ($P = 0.53$). No significant differences in overall IOP change were observed between TT (coefficient −1.2 mmHg; 95% CI −3.5 to 1.1, $P = 0.40$) or TF (0.3; −2.1 to 2.8, 0.82) compared to WT group. During the performance (from baseline to 101 seconds), IOP increased significantly in WT (3.1, 0.6 to 5.7, 0.007) and TT groups (2.8, 0.2 to 5.3, 0.02), while TF group did not show a significant increase. However, no significant differences were found when comparing among the three groups. After the performance (150 seconds), mean IOP differences between groups were small and were not significantly different.

**Data availability statement:** The data underlying the results presented in the study are available from the OSF database; https://osf.io/ds8za.

**Funding:** The author(s) received no specific funding for this work.

**Competing interests:** The authors have declared that no competing interests exist.

## Conclusions

The three types of wind instruments demonstrated no significant differences in IOP elevation during or after performance. While a transient increase in IOP was observed during playing in all groups, IOP returned to baseline within 30 seconds after the performance.

## Introduction

Increased intraocular pressure (IOP) is an important risk factor for glaucomatous optic nerve damage. The occurrence of optic nerve damage may be the result of a temporary increase in IOP during a Valsalva maneuver. Apart from various activities that trigger a Valsalva-like maneuver, playing a high-resistance wind instrument has been reported to increase IOP and can be a risk factor for glaucoma [1].

The wind instruments were categorized into high-resistance and low-resistance groups. High resistance wind instruments (HRWI), such as the trumpet and French horn, required greater breath pressure and lower airflow rates to produce sound. In contrast, low-resistance wind instruments, such as the saxophone and trombone, require less breath pressure and involve higher airflow rates.

A previous study reported a temporary but significant increase in IOP while playing both high-resistance and low-resistance Western wind instruments [2]. Moreover, the increase in IOP was higher among HRWI players, resulting in a slight but compelling incidence of visual field damage [3]. The uveal engorgement observed during the Valsalva maneuver, as detected by the ultrasound biomicroscope, may be associated with IOP elevation [3]. Furthermore, the magnitude of the IOP rise depended on the amount of expiratory resistance, tone frequency, duration, and playing intensity [3,4].

The magnitude of IOP increase is greater in HRWI players than in low-resistance wind players [2]. Western wind musical instruments tend to be played through a mouthpiece, resulting in high-resistance blowing. Thai wind musical instruments, on the other hand, lack mouthpieces, which allow for low-resistance blowing. Moreover, there are two types of Thai wind instruments: Thai traditional and Thai folk. Thai traditional wind instrument examined in this study is the Khlui [5], which is a flute-like instrument constructed from bamboo or hardwood. Thai folk wind instrument is the Khaen [6] comprising mouth organs with bamboo pipes. Since those who play Thai folk wind instruments need to store air in their cheeks and then push it out through the mouth to make a sound, higher IOP can be expected in these players compared to those who play Thai traditional wind instruments. To our knowledge, there are no reports of IOP changes while playing Thai wind instruments.

The objective of this study is to evaluate the changes in IOP before, during, and after playing Thai traditional, Thai folk, and Western wind instruments in healthy musicians.

## Materials and methods

This cross-sectional study was conducted at Srinagarind Hospital, Khon Kaen, Thailand, with the approval of the Khon Kaen University Ethics Committee for Human

Research (HE641149) in accordance with the tenets of the Declaration of Helsinki. This study is registered at www.thai-clinicaltrials.org (TCTR20211217001). The recruitment period was August 6th and 24th, 2022.

## Clinical examination

Fifteen healthy musicians, 21–30 years old, were recruited from students at the Faculty of Fine and Applied Arts, Khon Kaen University. Written informed consent was obtained from all participants. Participants were excluded if they had the following conditions: history of glaucoma, suspected glaucoma, uveitis, corneal haze or opacity preventing IOP measurement, retinal detachment, hypertension, asthma, or cardiovascular diseases. Demographic data and ocular examinations, including age, gender, best-corrected visual acuity (BCVA), IOP measuring by Goldmann applanation tonometry (GAT), central corneal thickness (CCT), body mass index (BMI), and gonioscopy were obtained to ensure they were deprived of any ocular diseases.

## Performance protocol and breathing techniques

The saxophone and trumpet were selected to represent the Western wind instrument (WT) group, the Khlui for the Thai traditional wind instrument (TT) group, and the Khaen for the Thai folk wind instrument (TF) group. Participants were assigned to each group based on their proficiency with the respective instrument. All participants received identical sets of musical notes, consisting of a repeated cycle of whole-step notes in the C major scale (Do to Ti and Ti to Do), as demonstrated in S1 Fig. The tempo was set so that each note lasted for one second, requiring each participant to play 120 notes within 120 seconds.

The breathing technique for each instrument differed. For the saxophone and trumpet, participants used diaphragmatic (abdominal) breathing, inhaling quickly and deeply through the mouth. The breath fills the lungs, causing the waist to expand outward, and air was exhaled steadily and controlled by the diaphragm to produce a consistent tone. For the Khlui, participants also employed diaphragmatic breathing, inhaling through the nose and controlling the airflow with the diaphragm. The breath is then controlled and released through the mouth to produce sound. There is no use of cheek air storage or circular breathing. In contrast, the Khaen required the use of circular breathing. Participants inhaled through the nose while simultaneously pushing stored air from the cheeks into the instrument, allowing for continuous sound production. All musicians were seated during both the performance and IOP measurement. Documentation of each instrument and the participants' postures across all groups is provided in S2 Fig. The individual pictured in S2 Fig has provided written informed consent (as outlined in PLOS consent form) to publish their image alongside the manuscript.

## IOP measurement

Before playing the instrument, IOP was measured in both eyes at zero second ($T_0$). When the instruments were being played, IOP was measured in the right eye at 37 seconds ($T_{37}$) and in the left eye at 101 seconds ($T_{101}$). These measurements were designed to be taken while the participants played the high notes. S1 Fig demonstrates the timepoint of IOP measurement in each eye. IOP of both eyes could not be measured at the same time during the performance because the instrument blocks the approach. Therefore, we decided to measure the IOP of each eye at different time points. The evaluator obtained the IOP while the participants performed continuously for 120 seconds. After playing the instrument, IOP was measured in both eyes at 150 seconds ($T_{150}$) (30 seconds after playing stopped) and at 240 seconds ($T_{240}$) (120 seconds after playing stopped). All IOP measurements were performed by a single evaluator using one Tono-Pen AVIA® (Reichert, Inc., Depew, NY). According to the device's measurement technique, the reported IOP values represent the average of five taps (or ten measurements) along with 95% confidence intervals. A demonstration of a participant from each group performing on their respective instrument during IOP measurement is presented in S2 Fig. CCT-adjusted IOP values were calculated using the online IOP calculator [7]. These corrected IOP values were subsequently used for the analysis. The primary outcome was the difference in IOP between groups while playing the

instruments (IOP at $T_{37}$ and IOP at $T_{101}$). Secondary outcomes were the difference in IOP between groups after the instruments stopped playing for 30 and 120 seconds (IOP at $T_{150}$ and IOP at $T_{240}$). The trend of IOP changes was also evaluated for the three groups.

## Sample size estimation

The sample size was estimated considering a power of more than 0.8 with an alpha error of 0.05. According to the linear mixed-effects model for repeated measures, fixed effects refer to the type of instrument and the timepoint of IOP measurement, while random effects refer to eye laterality (right or left) and subjects. We simulated the IOP data at each timepoint, gathered from previous studies. For Thai instruments, the data were obtained from Thai musicians. Based on the calculation using R (version 4.2.0, R Foundation for Statistical Computing, Vienna, Austria) [8], five participants were found to be adequate in each group to obtain a total of 120 datapoints: 30 datapoints at $T_0$, 15 datapoints at $T_{37}$, 15 datapoints at $T_{101}$, 30 datapoints at $T_{150}$, and 30 datapoints at $T_{240}$. Enrollment of 15 participants would provide a power of 0.88 to detect the IOP difference in the three groups.

## Statistical analysis

Mean and standard deviation, or the median and interquartile range (IQR), were used to describe the continuous variables according to their distribution. Percentage was used to describe the categorical variables. The Kruskal-Wallis test and Fisher's exact test were used for multiple comparisons. The linear mixed-effects model for repeated measures (MMRM) was used to assess the mean differences in IOP, a continuous variable, between groups. The analysis was adjusted for age and sex covariates. Fixed effects included group, timepoint (baseline, 37, 101, 150, 240 seconds), and baseline IOP. Additionally, second-matched baseline IOP and the interaction between timepoint and second-matched baseline IOP were included as covariates. A random intercept was used to account for potential inter-individual variability in baseline IOP between the right and left eyes. All statistical analyses were performed using R (version 4.2.0, R Foundation for Statistical Computing, Vienna, Austria) [8]. The lme4 package [9] was used for MMRM analysis. A *P*-value of <0.05 was considered statistically significant.

## Results

Thirty eyes from 15 wind instrument musicians were enrolled in the study, with a mean (SD) age of 23.3 (3.2) years. The mean (SD) duration of wind instrument experience was 3.9 (1.1) years, with no significant difference observed between groups (*P* = 0.35). Table 1 summarized the baseline characteristics of the musicians. The participants in the Western group were older than those in the other groups. CCT was higher in the Thai traditional group compared to the others, therefore, CCT-adjusted IOP value was calculated. Corrected IOP values of the right and left eyes did not differ significantly between groups, nor between eyes within any group. No significant differences were observed in other demographic or clinical characteristics among the groups. Gonioscopic examination confirmed that all participants had open anterior chamber angles. In the Western group, three of the five musicians played the saxophone (two females, one male), and two played the trumpet (one female, one male). The saxophone and trumpet players had mean ages of 24.6 and 26 years, respectively, and mean playing experience of 4.6 and 4.5 years.

Table 2 shows the mean IOP at each timepoint. The mean (SD) baseline IOP measured by Tono-Pen® was 13.3 (2.7), 12.3 (1.5), and 14.5 (2.4) mmHg for the Western (WT) group, Thai traditional (TT) group, and Thai folk (TF) group, respectively, with no significant difference between groups (*P* = 0.53). Thirty-seven seconds after the performance, IOP increased in all groups and continued to increase until the timepoint of 101 seconds. By fitting the linear mixed-effects (LME) model to the data on each participant, the spaghetti plot demonstrated a change in IOP throughout the performance of each participant in the three groups (Fig 1A).

**Table 1. Baseline characteristics of musicians playing the three types of wind instruments.**

| Characteristics | Wind instrument type | | | |
|---|---|---|---|---|
| | Western<br>n = 10 eyes | Thai traditional<br>n = 10 eyes | Thai folk<br>n = 10 eyes | *P*-value |
| Age; year, median (IQR) | 24 (22-28) | 21 (21-24) | 21 (20-23) | 0.02 |
| Gender; n (%) | | | | 0.08 |
| Male | 6 (60) | 6 (60) | 10 (100) | |
| Female | 4 (40) | 4 (40) | – | |
| BCVA in LogMAR; median (IQR), min–max | 0 (0−0), 0-0.17 | 0 (0−0), 0−0 | 0 (0−0), 0-0.17 | 0.60 |
| IOP[a]; mmHg, median (IQR) | | | | |
| Both eyes | 14<br>(12.2-15.5) | 11.5<br>(11-15) | 14<br>(13.2-14.8) | 0.53 |
| Right eye | 13<br>(12-14) | 12<br>(11-15) | 14<br>(14−14) | 0.76 |
| Left eye | 14<br>(14-16) | 11<br>(11-15) | 14<br>(13-15) | 0.51 |
| *P*-value[b] | 0.10 | 0.35 | >0.99 | |
| CCT; microns, median (IQR) | 538<br>(534-542) | 566<br>(544-610) | 540<br>(524-549) | 0.04 |
| Corrected IOP; mmHg, median (IQR) | | | | |
| Both eyes | 12.5<br>(11.1-13.4) | 10.2<br>(9.75-10.4) | 13.8<br>(12.1-15.6) | 0.01 |
| Right eye | 11.2<br>(11.1-12.1) | 10.2<br>(9.9-10.4) | 13.8<br>(12.4-15.2) | 0.11 |
| Left eye | 13<br>(12.8-13.6) | 10.2<br>(8.7-10.4) | 13.7<br>(12-15.8) | 0.05 |
| *P*-value[b] | 0.06 | 0.37 | 0.81 | |
| BMI; kg/m², median (IQR) | 23.7<br>(22.5-25.1) | 22.6<br>(20.7-23.4) | 23.4<br>(22.4-23.5) | 0.81 |
| Experience; years, median (IQR) | 5<br>(4-5) | 3<br>(3-5) | 3<br>(2-5) | 0.35 |

Abbreviations: IQR, interquartile range; BCVA, best-corrected visual acuity; IOP, intraocular pressure; CCT, central corneal thickness; BMI, body mass index.

[a]Measured using the Goldmann applanation tonometer.

[b]*P*-value of right and left eyes in the same group by Wilcoxon signed rank test.

## IOP difference between groups

When compared to the WT group, the TT group showed no statistical difference in overall IOP change (coefficient −1.2; 95% CI −3.5 to 1.1, *P* = 0.40). Similarly, the TF group showed no statistical difference in overall IOP change compared to the WT group (coefficient 0.3; 95% CI −2.1 to 2.8, *P* = 0.82). Fig 1B presents the IOP change throughout the performance in each group after fitting the model.

## IOP changes during the performance

From baseline to 37 seconds, IOP increased by 2.5 mmHg (95% CI −0.1 to 5.1, *P* = 0.06), 2.4 mmHg (−0.1 to 5.0, 0.07), and 1.8 mmHg (−0.7 to 4.4, 0.47) in the WT, TT, and TF groups, respectively, corresponding to increases of 18.8%, 19.5%, and 12.4% (Table 2). Although the TF group revealed slightly lower IOP increases at $T_{37}$ and $T_{101}$ compared to the other two groups, these differences were not statistically significant (*P* = 0.51 and 0.37, respectively).

**Table 2. Mean intraocular pressures and the intraocular pressure changes in musicians playing the three types of wind instruments at each specified timepoint.**

| Timepoint | Wind instrument type | | | | | | |
|---|---|---|---|---|---|---|---|
| | Western | | Thai traditional | | Thai folk | | |
| | Mean (SD) | IOP change from previous timepoint (%) | Mean (SD) | IOP change from previous timepoint (%) | Mean (SD) | IOP change from previous timepoint (%) | *P*-value |
| $T_0$ (before playing) | 13.3 (2.7) | | 12.3 (1.5) | | 14.5 (2.4) | | 0.53 |
| $T_{37}$ (37 sec. after playing) | 15.8 (2.6) | 18.8 | 14.7 (2) | 19.5 | 16.3 (1.1) | 12.4 | 0.51 |
| $T_{101}$ (101 sec. after playing) | 16.4 (0.9) | 3.8 | 15.1 (3.1) | 2.7 | 16.9 (1.7) | 3.7 | 0.37 |
| $T_{150}$ (30 sec. after stop playing) | 13.1 (2.4) | −20.1 | 13 (2.8) | −13.9 | 13.6 (2.1) | −19.5 | 0.92 |
| $T_{240}$ (120 sec. after stop playing) | 12.7 (2.2) | −3.1 | 11 (1.5) | −15.4 | 13.7 (1.7) | 0.7 | 0.12 |

Abbreviations: SD, standard deviation; IOP, intraocular pressure.

Adjusted by age and sex.

We emphasize evaluating the peak IOP at $T_{101}$ since this might have a greater effect on the optic nerve compared to the lower IOP at $T_{37}$. From baseline to 101 seconds, the IOP increased significantly in the WT and TT groups by 3.1 mmHg (0.6 to 5.7, 0.007) and 2.8 mmHg (0.2 to 5.3, 0.02), respectively. In contrast, the IOP in the TF group did not show a statistically significant increase, showing a change of 2.4 mmHg (−0.2 to 4.9, 0.09). This resulted in an increase of 23.3%, 22.8%, and 16.6% in IOP for the three groups. The mean differences (mmHg) in increased IOP between WT vs. TT (0.3), WT vs. TF (0.7), and TT vs. TF (0.4) were not significant during the performance. Table 3 shows the mean IOP differences during the performance of each group.

### IOP changes after the performance

At thirty seconds after the performance ($T_{150}$), the WT group showed mean IOP differences of 2.3 mmHg (95%CI −5.4 to 6.0) and 0.4 mmHg (−5.9 to 6.6) compared to the TT and TF groups, respectively. The TT group showed a mean difference of 0.07 mmHg (−5.8 to 6.0) compared to the TF group. Additionally, at 120 seconds after the performance ($T_{240}$), the TF group demonstrated mean differences of 2.0 mmHg (−3.9 to 7.9) and 0.1 mmHg (−6.1 to 6.4) compared to the TT and WT groups, respectively, while the WT group had a mean difference of 1.9 (−3.8 to 7.6) compared to the TT group. Statistical analysis revealed no significant differences in mean IOP between groups at this timepoint.

From 101 seconds (peak IOP) to 240 seconds, mean IOP significantly decreased in all groups, as demonstrated in Table 3. Specifically, reductions of 22.6%, 27.2%, and 18.9% were observed in the WT, TT, and TF groups, respectively. Comparisons between groups confirmed no significant difference in the mean IOP after the performance.

## Discussion

This study demonstrates that IOP significantly increased during the playing of Thai traditional and Western wind instruments. However, there were no statistical differences in IOP elevation between groups during the playing of three types of wind instruments, and these IOP elevations decreased to the baseline value within 30 seconds once the participants had stopped playing the instruments. To our knowledge, this appears to be a relatively unexplored area, particularly regarding IOP changes associated with playing Thai traditional and Thai folk wind instruments.

The Khaen (Thai folk wind instrument) is a mouth organ made from bamboo pipes of many different sizes connected to a central hollowed-out hardwood reservoir. Each bamboo pipe contains small metal free reeds of different thicknesses to produce different notes. Players must store air in their cheeks and then blow into the reservoir via a small hole, which is a part of the hardwood reservoir. The Khlui (Thai traditional wind instrument) is a flute-like instrument made from bamboo,

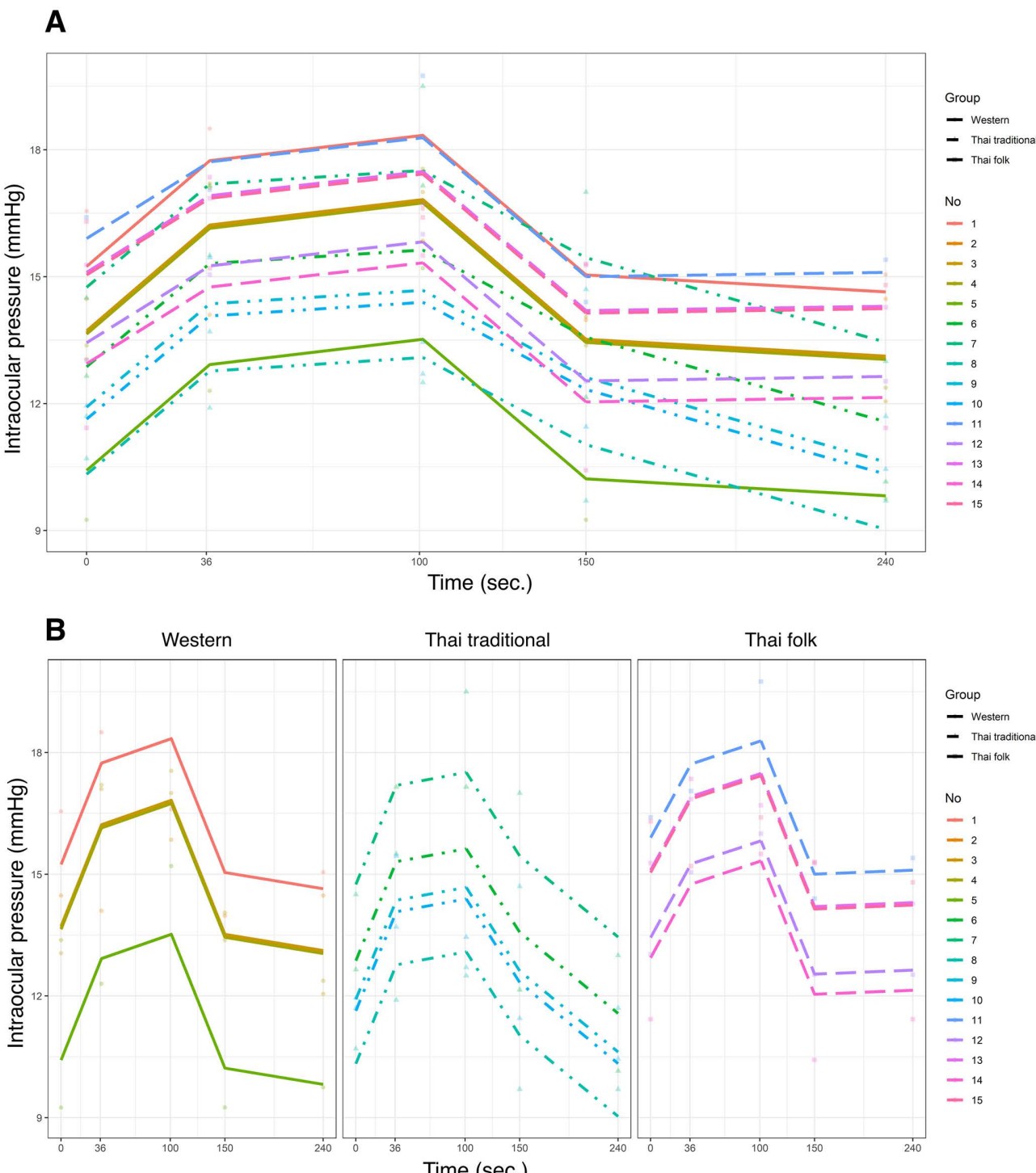

**Fig 1. Spaghetti plots showing the intraocular pressure changes in each patient.** Intraocular pressure changes in each patient for overall comparison (A) and between-group comparison (B) during playing (0 to 101 seconds) and after playing (150 to 240 seconds) the wind instruments. The IOP trends of participants No. 2, 3 and 4 (in the WT group) are plotted in the same line. The IOP trends of participants No.13 and 15 (in the TF group) are plotted in the same line.

**Table 3. Mean IOP differences in musicians during and after playing the three types of wind instruments.**

| | During performance (IOP at $T_{101}$ − IOP at $T_0$) | | | After performance (IOP at $T_{240}$ − IOP at $T_{101}$) | | |
|---|---|---|---|---|---|---|
| | estimate | 95% CI | *P*-value | estimate | 95% CI | *P*-value |
| Group | IOP differences between timepoints for each wind instrument type (mmHg) | | | | | |
| WT | 3.1 | 0.6, 5.7 | 0.007 | −3.7 | −6.3, −1.2 | 0.0007 |
| TT | 2.8 | 0.2, 5.3 | 0.02 | −4.1 | −6.6, −1.5 | 0.0002 |
| TF | 2.4 | −0.2, 4.9 | 0.09 | −3.2 | −5.7, −0.6 | 0.005 |
| Group | IOP differences between types of wind instrument (mmHg) | | | | | |
| WT vs TT | 0.3 | −4.1, 4.7 | >0.999 | 0.4 | −4.0, 4.8 | >0.999 |
| WT vs TF | 0.7 | −3.7, 5.1 | >0.999 | −0.5 | −5.0, 3.9 | >0.999 |
| TT vs TF | 0.4 | −4.0, 4.8 | >0.999 | −0.9 | −5.3, 3.5 | >0.999 |

Abbreviations: IOP, intraocular pressure; WT, Western wind instrument; TT, Thai traditional wind instrument; TF, Thai folk wind instrument; MD, mean difference.

Adjusted by age and sex.

with eight finger holes that help produce different notes. Musicians must blow air into the small hole at the end of the instrument to produce sound. Since there is no metal inside the Khlui, the sound-making mechanisms differ between these Thai wind instruments. We observed that players used more blow force when playing the Khaen compared to the Khlui due to the larger air reservoir, so we expected higher IOP rises in Khaen players during the performance.

Thai wind instruments showed a similar trend in IOP changes compared to Western wind instruments, although the musicians had to blow air through the instruments, applying various efforts and techniques. Moreover, IOP in Thai folk players who had to store air in their cheeks and push it out was not different compared to IOP in Thai traditional players. This result was similar to a previous study reporting nonsignificant IOP difference between high-resistance and low-resistance wind instruments [10]. It can be concluded that Thai wind instruments cause a temporary IOP rising comparable to that observed with Western wind instruments, despite differences in playing and breathing techniques.

A previous study on Western wind instruments reported an 11 mmHg in IOP change after the participants played the saxophone at a high-pitched and loud level for 12 seconds [3]. A greater change in IOP [2,3] and high central retinal venous pressure [11] were demonstrated in trumpet players. In our study, an IOP change at 37 seconds during the performance was lower than 6 mmHg in both saxophone and trumpet players. We assumed that the use of a low tone frequency, low playing intensity, and an identical set of musical notations in our study might be the cause of the lower IOP change within each group. These low-intensity settings might also explain the nonsignificant differences in IOP elevation between groups. Further studies employing higher tone frequencies and increased playing intensity might help determine whether these factors affect IOP changes across different instrument groups.

Other factors that might affect the IOP include participants' overweight status and sex. Obesity has been identified as an independent risk factor for increased IOP in both men and women [12]. In our study, although one participant in each group was overweight according to the body mass index (BMI > 25 kg/m²), the baseline BMI was not different among groups. Additionally, the IOP measurements during performance (at $T_{37}$ and $T_{101}$) in these participants remained below 21 mmHg. Notably, the TF group consisted entirely of male participants, whereas the other groups included both genders. While no statistically significant difference in sex distribution was observed among groups ($P = 0.08$), potential sex-related effects on IOP warrant cautious interpretation [13]. Therefore, sex was included as a covariate in the statistical analysis to adjust for its potential confounding effect.

After the performance, IOP decreased and returned to baseline within 30 seconds in all groups, followed by a plateau or slight decrease. This result was similar to a previous study [4] that demonstrated a decrease in IOP to the baseline

value 40 seconds after the participants had stopped playing brass instruments and 20 seconds after playing woodwind instruments. Since the musicians in our study were young and healthy, it is likely that their ocular autoregulation (the capacity to maintain the ocular blood flow when the IOP is high [14]) may be better in young adolescents than in the older. We assume that this rapid return of IOP after the performance may be less efficient in older musicians or those with glaucoma. This is likely due to ischemic optic nerve damage associated with aging and glaucoma, which can impair ocular autoregulation [15]. Consequently, these individuals may experience delayed or inadequate IOP normalization following elevated IOP episodes.

A previous study has shown that glaucoma patients exhibited higher mean IOP and greater IOP fluctuation after playing the high- and low-resistance wind instruments for 20 minutes, compared to individuals without glaucoma [10]. One case report described a glaucoma patient who experienced a markedly increase in IOP while playing the oboe. The IOP elevation was more pronounce in the eye without prior trabeculectomy, whereas the eye with advanced glaucoma and a functional trabeculectomy showed a smaller increase. Notably, IOP returned to baseline levels within five minutes of rest [16]. Although our study was conducted in young, healthy musicians, we hypothesize that the transient IOP elevation may be more pronounced in musicians with pre-existing glaucoma or ocular hypertension, potentially due to impaired ocular autoregulation. This transient elevation could contribute to an increased risk of glaucoma progression in such individuals.

The strength of this study is the measurement of IOP in Thai wind instruments, which has not previously been reported. Using an identical set of musical notations is important for minimizing the variation in notes from each instrument. We presumed that the highest IOP change could be detected when measured at the highest frequency tone, so IOP was measured when the musician played on the same high musical note at each different timepoint.

There are limitations to this study that should be acknowledged. First, IOP measurement was performed using the Tono-Pen, as the use of Thai instruments obstructed the slit-lamp examination and precluded the application of Goldmann applanation tonometry. It is important to note that the measurement errors may have occurred due to dynamic conditions during the performance period. Nevertheless, assessing ten consecutive IOP measurements with the Tono-Pen, as described, likely reduce intra-observer variability. Second, there were two types of instruments in the Western wind instrument group. We tried to include participants of the same age, but that resulted in a limited number of musicians who played the same instrument. Since the study objective was to compare the IOP in Western and Thai wind instrument groups, due to their different materials and sound-making mechanisms, we decided to include the saxophone and trumpet, which were both made from brass, in the same group of instruments. Third, the performance period in our study was relatively short. In reality, many Thai songs last longer than two minutes, hence, a greater rise in IOP during the playing of the instruments might be expected. Additionally, the sample size of five participants per group (total n = 15) is relatively small. Although this sample size achieved a statistical power of 0.88 to detect differences in IOP among the three groups, caution is warranted because this limited number might not fully capture inter-subject variability. Furthermore, there is a possibility that the effect size from the simulated data that used for power calculation was overestimated, potentially leading to an underestimation of the required sample size. Lastly, the result of this study cannot be extrapolated to patients with normal-tension glaucoma, high-tension glaucoma, or glaucoma suspects whom the IOP fluctuation during the performance might be higher. Further research involving longer performance time, higher frequency notes, larger sample sizes, and the inclusion of normal and glaucoma participants is recommended to explore potential IOP elevation during the playing of Western and Thai wind instruments.

## Conclusion

No significant differences in IOP were observed before, during or after playing Thai traditional, Thai folk, or Western wind instruments. Although a mild and transient elevation in IOP was noted during performance, IOP levels returned to baseline within 30 seconds after performance was stopped in all groups. These findings suggest that playing Thai and Western wind instruments for a short period of time should not be a concern for increasing the risk of glaucoma in young, healthy

individuals. Nevertheless, caution should be warranted when interpreting these results in individuals with glaucoma or glaucoma suspects.

## Supporting information

**S1 Fig. The musical scale used by all musicians and the timepoint of IOP measurement in each eye.**
(PDF)

**S2 Fig. The instruments used in each group and the participants' postures during IOP measurement across all group.**
(PDF)

## Acknowledgments

The authors would like to acknowledge the efforts of the musicians from the Faculty of Fine and Applied Arts, Khon Kaen University, Khon Kaen, Thailand.

## Author contributions

**Conceptualization:** Jirach Jatechayanon, Somkiat Asawaphureekorn.

**Data curation:** Piyanan Suparattanagool, Sukhumal Thanapaisal.

**Formal analysis:** Somkiat Asawaphureekorn, Piyanan Suparattanagool.

**Investigation:** Jirach Jatechayanon, Sukhumal Thanapaisal.

**Methodology:** Jirach Jatechayanon, Somkiat Asawaphureekorn, Piyanan Suparattanagool, Sukhumal Thanapaisal.

**Project administration:** Jirach Jatechayanon, Sukhumal Thanapaisal.

**Software:** Piyanan Suparattanagool.

**Supervision:** Somkiat Asawaphureekorn, Sukhumal Thanapaisal.

**Validation:** Somkiat Asawaphureekorn, Piyanan Suparattanagool, Sukhumal Thanapaisal.

**Visualization:** Sukhumal Thanapaisal.

**Writing – original draft:** Jirach Jatechayanon.

**Writing – review & editing:** Somkiat Asawaphureekorn, Sukhumal Thanapaisal.

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
