## [Decision Letter · Decision Letter 0]

16 May 2025

PONE-D-25-12979Changes in Intraocular Pressure Before, During, and After Playing Thai Traditional, Thai Folk, and Western Wind InstrumentsPLOS ONE

Dear Dr. Thanapaisal,

Thank you for submitting your manuscript to PLOS ONE. After careful consideration, we feel that it has merit but does not fully meet PLOS ONE’s publication criteria as it currently stands. Therefore, we invite you to submit a revised version of the manuscript that addresses the points raised during the review process.

Please submit a revised manuscript that address issues raised by the reviewers, particularly addressing confounding, sample size as a limitation and providing more details as requested by Reviewer 2.

We look forward to receiving your revised manuscript.

Kind regards,

Jae Hee Kang

Academic Editor

PLOS ONE

Reviewers' comments:

Reviewer's Responses to Questions

**Comments to the Author**

1. Is the manuscript technically sound, and do the data support the conclusions?

Reviewer #1: Yes

Reviewer #2: Partly

2. Has the statistical analysis been performed appropriately and rigorously? 

Reviewer #1: Yes

Reviewer #2: Yes

3. Have the authors made all data underlying the findings in their manuscript fully available?

Reviewer #1: Yes

Reviewer #2: Yes

4. Is the manuscript presented in an intelligible fashion and written in standard English?

Reviewer #1: Yes

Reviewer #2: Yes

5. Review Comments to the Author

Reviewer #1: Overall, this was a well-executed study.

A major comment is that the authors did not adjust for sex and age differences in their analyses; please re-do the analyses adjusting for these factors.

Another major comment would be that the authors should acknowledge that the results being non-significant could be because the sample size was too small.

Reviewer #2: This cross-sectional study evaluates intraocular pressure (IOP) changes before, during, and after playing Thai traditional (Khlui), Thai folk (Khaen), and Western (saxophone/trumpet) wind instruments in 15 healthy young musicians. Using mixed-effects models, the authors report significant transient IOP elevations during performance across all groups, with no inter-instrument differences, and rapid return to baseline within 30 seconds post-performance.

Comments

1. As a novel study, it adds valuable knowledge to the field.

2. The use of identical notes and standard duration effectively minimizes performance variability between groups.

3. The statistical approach using mixed-effects models is sound.

4. The research has good clinical relevance to both musicians and eye practitioners.

Concerns

1. No information is provided about players' experience or skill levels, which may influence blowing technique and IOP changes.

2. While using Tono-Pen during performance is understandable, the number of IOP measurements taken at each timepoint is unclear. Was there only one measurement per timepoint? The measurement error under dynamic conditions requires discussion, including intra-observer variability and comparison with standard methods e.g., GAT.

3. The player position (sitting/standing) during performance and IOP measurement is not specified, despite the known effect of posture on IOP. Address how breathing technique and posture differences between groups might influence IOP changes.

4. IOP measurements from right and left eyes should be separately analyzed. The percentage change at 37s should reference baseline IOP for right eyes, while change at 101s should reference baseline IOP for left eyes. Table 1 implicitly assumes equal mean IOP for both eyes in each group, which although possible, is unlikely.

5. The terms "high-resistance" and "low-resistance" wind instruments need clearer definition for readers unfamiliar with wind-instrument acoustics.

6. Images of the Khlui and Khaen would help readers understand these unfamiliar instruments. Better still, an image of a participant in each of the groups blowing their instrument while IOP is being recorded would be a valuable addition to the methods section.

7. Combining saxophone and trumpet players into a single "Western" group is problematic given their different blowing techniques and resistance levels. Although this limitation was addressed in the discussion, data on specific Western instruments played would be valuable (e.g., how many people played the sax vs trumpet).

8. Expand discussion of clinical implications, particularly for musicians with pre-existing glaucoma or ocular hypertension.

6. PLOS authors have the option to publish the peer review history of their article (what does this mean? ). If published, this will include your full peer review and any attached files.

**Do you want your identity to be public for this peer review?** For information about this choice, including consent withdrawal, please see our Privacy Policy .

Reviewer #1: No

Reviewer #2: No

---

## [Author Response · Author response to Decision Letter 1]

27 Jun 2025

** All of below are attached as a response to reviewer file.**

Dear Reviewers,

Thank you for taking time to read and review our manuscript, which has greatly contributed to enhancing its scientific quality.

We have revised the manuscript and provided point-by-point responses as follow:

Reviewer #1

Response from reviewer:

1. The authors did not adjust for sex and age differences in their analyses; please re-do the analyses adjusting for these factors.

New/edit sentence in manuscript:

All the results in the manuscript text, Table 2 and Table 3 were updated.

Line 154

“The analysis was adjusted for age and sex covariates.”

Explanation:

Thank you for your suggestion. We re-analyzed the data with adjustments for age and sex and found that the main outcome remained consistent with our previous report. The results in the manuscript text and all tables have been revised accordingly.

Response from reviewer:

2. The authors should acknowledge that the results being non-significant could be because the sample size was too small.

New/edit sentence in manuscript: -

Explanation:

Thank you for your concern regarding the sample size. As detailed in the Methods section, we conducted a sample size calculation based on mixed-effects model analysis using five repeated IOP measurements from both eyes. The estimated required sample size was five patients per group. Although the number of participants may appear small, all subjects contributed 120 data points in total. Based on this model, the calculated statistical power was 88%, with an alpha error of 0.05, as reported in the manuscript. Furthermore, we have provided a document file demonstrating the sample size calculation using the R programming language, which includes details on the fixed and random effects applied to simulated data. In light of this, we believe the sample size is sufficient to support the validity of our analysis. We appreciate your thoughtful consideration.

Reviewer #2

Response from reviewer:

No information is provided about players’ experience or skill levels, which may influence blowing technique and IOP changes.

New/edit sentence in manuscript

Line 168

“The mean (SD) duration of wind instrument experience was 3.9 (1.1) years, with no significant difference observed between groups (P=0.35).”

Please see Table 1 – Experience, years

Explanation:

Thank you for your valuable comments. We have included information on the musicians’ experience by reporting the mean number of years of instrument practice, based on participants’ self-report. Additionally, we have provided the years of experience for participants in each group in the revised Table 1.

Response from reviewer:

2. While using Tono-Pen during performance is understandable, the number of IOP measurements taken at each timepoint is unclear. Was there only one measurement per timepoint? The measurement error under dynamic conditions requires discussion, including intra-observer variability and comparison with standard methods e.g., GAT.

New/edit sentence in manuscript

Line 125

“All IOP measurements were performed by a single evaluator using one Tono-Pen AVIA® (Reichert, Inc., Depew, NY). According to the device’s measurement technique, the average IOP with its 95% confidence interval were calculated and displayed after five taps (or ten measurements).”

Line 334

“Firstly, IOP measurement was performed using the Tono-Pen, as the use of Thai instruments obstructed the Slit-lamp examination and precluded the application of Goldmann applanation tonometry. It is important to note that the measurement errors may have occurred due to dynamic conditions during the performance period. Nevertheless, assessing ten consecutive IOP measurements with the Tono-Pen, as described, likely reduce intra-observer variability.”

Explanation

We have added details on the number of IOP measurements and the use of the Tono-Pen in the Methods section, along with a brief discussion of its limitations.

Response from reviewer:

3. The player position (sitting/standing) during performance and IOP measurement is not specified, despite the known effect of posture on IOP. Address how breathing technique and posture differences between groups might influence IOP changes.

New/edit sentence in manuscript

Line 101

“The breathing technique for each instrument differed. For the saxophone and trumpet, participants used diaphragmatic (abdominal) breathing, inhaling quickly and deeply through the mouth. The breath fills the lungs, causing the waist to expand outward, and air was exhaled steadily and controlled by the diaphragm to produce a consistent tone. For the Khlui, participants also employed diaphragmatic breathing, inhaling through the nose and controlling the airflow with the diaphragm. The breath is then controlled and released through the mouth to produce sound. There is no use of cheek air storage or circular breathing. In contrast, the Khaen required the use of circular breathing. Participants inhaled through the nose while simultaneously pushing stored air from the cheeks into the instrument, allowing for continuous sound production. All musicians were seated during both the performance and IOP measurement. Documentation of each instrument and the participants’ postures across all groups is provided in S2 Fig.”

Explanation

We added a paragraph to address the breathing techniques associated with each instrument and the posture of the musicians. Supplementary document S2 has been attached to demonstrate the musicians’ postures and the procedure used to obtain IOP measurements during performance.

Response from reviewer:

4. IOP measurements from right and left eyes should be separately analyzed. The percentage change at 37s should reference baseline IOP for right eyes, while change at 101s should reference baseline IOP for left eyes. Table 1 implicitly assumes equal mean IOP for both eyes in each group, which although possible, is unlikely.

New/edit sentence in manuscript

Line 154

“Fixed effects included group, timepoint (baseline, 37, 101, 150, 240 seconds), and baseline IOP. Additionally, second-matched baseline IOP and the interaction between timepoint and second-matched baseline IOP were included as covariates. A random intercept was used to account for potential inter-individual variability in baseline IOP between the right and left eyes.”

Line 174

“Additionally, IOP measurements did not differ significantly between the right and left eyes within any group.”

*Please also see Table 1*

Explanation

Thank you for your comment. We selected a mixed-effects model for this analysis. In accordance with the principles of mixed model analysis, intraocular pressures (IOPs) from both the right and left eyes of each individual were included, with the model accounting for within-subject correlation by specifying the eyes as random effects. We also employed a random intercept to account for potential differences in baseline IOP between individuals. We believe this method is appropriate and statistically justified. In response to reviewer’s suggestion, we’ve added the IOP data for right and left eyes of participants in each group, as presented in Table 1 and revised the paragraph in the statistical analysis subsection to explain more about the fixed effect and random intercept that we used in the analysis.

Response from reviewer:

5. The terms "high-resistance" and "low-resistance" wind instruments need clearer definition for readers unfamiliar with wind-instrument acoustics.

New/edit sentence in manuscript

Line 47

“The wind instruments were categorized into high-resistance and low-resistance groups. High resistance wind instruments (HRWI), such as the trumpet and French horn, required greater breath pressure and lower airflow rates to produce sound. In contrast, low-resistance wind instruments, such as the saxophone and trombone, require less breath pressure and involve higher airflow rates.”

Explanation

We added a paragraph in the introduction section to explain the difference between HRWI and LRWI.

Response from reviewer:

6. Images of the Khlui and Khaen would help readers understand these unfamiliar instruments. Better still, an image of a participant in each of the groups blowing their instrument while IOP is being recorded would be a valuable addition to the methods section.

New/edit sentence in manuscript

Line 112

“Documentation of each instrument and the participants’ postures across all groups is provided in S2 Fig.”

Line 134

“A demonstration of a participant from each group performing on their respective instrument during IOP measurement is presented in S2 Fig.”

Explanation

Thank you for this valuable comment. We have attached a document in the method section that demonstrates the instruments and the participant’s posture while IOP is being recorded.

Response from reviewer:

7. Combining saxophone and trumpet players into a single "Western" group is problematic given their different blowing techniques and resistance levels. Although this limitation was addressed in the discussion, data on specific Western instruments played would be valuable (e.g., how many people played the sax vs trumpet).

New/edit sentence in manuscript

Line 176

“In the Western group, three of the five musicians played the saxophone (two females, one male), and two played the trumpet (one female, one male). The saxophone and trumpet players had mean ages of 24.6 and 26 years, respectively, and mean playing experience of 4.6 and 4.5 years.”

Explanation

Thank you for the suggestion. We have added details of the participants playing saxophone and trumpet in the result section.

Response from reviewer:

8. Expand discussion of clinical implications, particularly for musicians with pre-existing glaucoma or ocular hypertension.

New/edit sentence in manuscript

Line 316

“A previous study has shown that glaucoma patients exhibited higher mean IOP and greater IOP fluctuation after playing the high- and low-resistance wind instruments for 20 minutes, compared to individuals without glaucoma9. One case report described a glaucoma patient who experienced a markedly increase in IOP while playing the oboe. The IOP elevation was more pronounce in the eye without prior trabeculectomy, whereas the eye with advanced glaucoma and a functional trabeculectomy showed a smaller increase. Notably, IOP returned to baseline levels within five minutes of rest. Although our study was conducted in young, healthy musicians, we hypothesize that the transient IOP elevation may be more pronounced in musicians with pre-existing glaucoma or ocular hypertension, potentially due to impaired ocular autoregulation. This transient elevation could contribute to an increased risk of glaucoma progression in such individuals.”

Explanation

Thank you for the insightful comment. We added paragraph in the discussion section.

---

## [Decision Letter · Decision Letter 1]

12 Jul 2025

PONE-D-25-12979R1Changes in Intraocular Pressure Before, During, and After Playing Thai Traditional, Thai Folk, and Western Wind InstrumentsPLOS ONE

Dear Dr. Thanapaisal,

Thank you for submitting your manuscript to PLOS ONE. After careful consideration, we feel that it has merit but does not fully meet PLOS ONE’s publication criteria as it currently stands. Therefore, we invite you to submit a revised version of the manuscript that addresses the points raised during the review process. The revision shows that the authors tried to address the previous reviewers' comments. Among the comments for the revised version, please address the need to acknowledge the various limitations provided by reviewers and conduct another set of analyses of IOP adjusted for CCT (or if not possible, address the limitations of not being able to do this).

We look forward to receiving your revised manuscript.

Kind regards,

Jae Hee Kang

Academic Editor

PLOS ONE

Journal Requirements:

Reviewers' comments:

Reviewer's Responses to Questions

**Comments to the Author**

1. If the authors have adequately addressed your comments raised in a previous round of review and you feel that this manuscript is now acceptable for publication, you may indicate that here to bypass the “Comments to the Author” section, enter your conflict of interest statement in the “Confidential to Editor” section, and submit your "Accept" recommendation.

Reviewer #3: All comments have been addressed

Reviewer #4: (No Response)

Reviewer #5: (No Response)

Reviewer #6: (No Response)

2. Is the manuscript technically sound, and do the data support the conclusions?

Reviewer #3: Yes

Reviewer #4: Partly

Reviewer #5: Partly

Reviewer #6: Partly

3. Has the statistical analysis been performed appropriately and rigorously? 

Reviewer #3: Yes

Reviewer #4: Yes

Reviewer #5: Yes

Reviewer #6: Yes

4. Have the authors made all data underlying the findings in their manuscript fully available?

Reviewer #3: Yes

Reviewer #4: Yes

Reviewer #5: Yes

Reviewer #6: Yes

5. Is the manuscript presented in an intelligible fashion and written in standard English?

Reviewer #3: Yes

Reviewer #4: Yes

Reviewer #5: Yes

Reviewer #6: Yes

6. Review Comments to the Author

Reviewer #3: Thank you for addressing previous reviewer's comments. The only thing I would ask you to consider adding to your conclusion (which you have in the discussion) is the caution needed with glaucoma patients or glaucoma suspects.

Reviewer #4: 1、The use of a linear mixed-effects model and simulation-based sample size estimation is methodologically appropriate, the inclusion of only 15 participants may raise concerns regarding the stability and generalizability of the model estimates. In models with multiple timepoints, fixed and random effects, and potentially correlated within-subject measures, a larger number of subjects is typically advisable to ensure robust variance estimation and avoid overfitting.

2、Discussion section, paragraph 1, In page no. 15 line no. 263-265: The claim of being the first study in this area should be made with caution. A more cautious phrasing, such as “this appears to be a relatively unexplored area…,” would be more appropriate and academically balanced.

Reviewer #5: While the potential for intraocular pressure (IOP) elevation during the use of wind instruments has been previously reported, this study is noteworthy for its relatively well-designed experimental setup and analytical approach, allowing for direct measurement and comparison of IOP during the playing of various wind instruments.

However, it would be advisable to ensure consistency between the Results and Conclusions sections of the abstract. If the conclusion states that “Elevated IOP was demonstrated during the playing of three types of wind instruments,” then this finding should also be explicitly presented in the Results section. Furthermore, since the primary outcome was the difference in IOP between the groups, it would enhance clarity to indicate in the abstract whether there were statistically significant differences in baseline IOP among the three groups.

Although the sample size was calculated to include five participants per group, based on a power analysis, a total of 15 participants is a relatively small sample size. This raises concerns that inter-subject variability may not be adequately captured. It is also possible that the effect size used in the simulation was overestimated, leading to an underestimation of the required sample size. These limitations should be acknowledged in the manuscript to allow readers to appropriately interpret the findings.

Reviewer #6: 1.Due to the consideration of gender as a factor affecting IOP changes during performance,the author should be recommended to explain whether baseline data will impact the study results in the Thai folk group, where all performers are male, compared to the other two groups.frequency

2.The authors should acknowledge that the results being non-significant could be because participants played the low tone frequency and low playing intensity musical notations, and further supplement studies data with higher strength notes .

3.The authors did not adjust for the effect of CCT on IOP in their analyses; please redo the analyses after Correction value .

7. PLOS authors have the option to publish the peer review history of their article (what does this mean? ). If published, this will include your full peer review and any attached files.

**Do you want your identity to be public for this peer review?** For information about this choice, including consent withdrawal, please see our Privacy Policy .

Reviewer #3: No

Reviewer #4: No

Reviewer #5: No

Reviewer #6: No

---

## [Author Response · Author response to Decision Letter 2]

14 Aug 2025

Please also find the attached 'response to reviewers' document.

Dear Reviewers,

Thank you for taking time to read and review our manuscript, which has greatly contributed to enhancing its scientific quality. We have revised the manuscript and provided point-by-point responses as follow:

Reviewer #3:

Response from reviewer

1. The only thing I would ask you to consider adding to your conclusion (which you have in the discussion) is the caution needed with glaucoma patients or glaucoma suspects.

New/edit sentence in manuscript

Line 369

“normal-tension glaucoma, high-tension glaucoma, or glaucoma suspects”

Line 382

“Nevertheless, caution should be warranted when interpreting these results in individuals with glaucoma or glaucoma suspects.”

Explanation

Thank you for your suggestion. I have added the sentence at the end of the conclusion paragraph. Also, I’ve added the ‘glaucoma suspects’ in the limitation of the discussion part.

Reviewer #4:

Response from reviewer

1. The use of a linear mixed-effects model and simulation-based sample size estimation is methodologically appropriate, the inclusion of only 15 participants may raise concerns regarding the stability and generalizability of the model estimates. In models with multiple timepoints, fixed and random effects, and potentially correlated within-subject measures, a larger number of subjects is typically advisable to ensure robust variance estimation and avoid overfitting.

New/edit sentence in manuscript

Line 362

“Additionally, the sample size of five participants per group (total n =15) is relatively small. Although this sample size achieved a statistical power of 0.88 to detect differences in IOP among the three groups, caution is warranted because this limited number might not fully capture inter-subject variability. Furthermore, there is a possibility that the effect size from the simulated data that used for power calculation was overestimated, potentially leading to an underestimation of the required sample size”

Line 372

“larger sample sizes”

Explanation

Thank you for your valuable comment. We have incorporated a statement regarding the sample size caution in the limitations section of the manuscript and also added the suggestion for the future study.

Response from reviewer

2. Discussion section, paragraph 1, In page no. 15 line no. 263-265: The claim of being the first study in this area should be made with caution. A more cautious phrasing, such as “this appears to be a relatively unexplored area…,” would be more appropriate and academically balanced.

New/edit sentence in manuscript

Line 271

“To our knowledge, this appears to be a relatively unexplored area, particularly regarding IOP changes associated with playing Thai traditional and Thai folk wind instruments.”

Explanation

Thank you so much. We have revised the sentence as you suggested.

Reviewer #5:

Response from reviewer

1. It would be advisable to ensure consistency between the Results and Conclusions sections of the abstract. If the conclusion states that “Elevated IOP was demonstrated during the playing of three types of wind instruments,” then this finding should also be explicitly presented in the Results section. Furthermore, since the primary outcome was the difference in IOP between the groups, it would enhance clarity to indicate in the abstract whether there were statistically significant differences in baseline IOP among the three groups.

New/edit sentence in manuscript

Line 30 in abstract

Thirty eyes from 15 participants were included, with mean (SD) baseline IOP of 13.3 (2.7), 12.3 (1.5), and 14.5 (2.4) mmHg in WT, TT, and TF groups (P = 0.53).

Line 34 in abstract

During the performance (from baseline to 101 seconds), IOP increased significantly in WT (3.1, 0.6 to 5.7, 0.007) and TT groups (2.8, 0.2 to 5.3, 0.02), while TF group did not show a significant increase. However, no significant differences were found when comparing among the three groups. After the performance (150 seconds), mean IOP differences between groups were small and did not show statistically significant.

Line 40 in abstract

“Conclusions: The three types of wind instruments demonstrated no significant differences in IOP elevation during or after performance. While a transient increase in IOP was observed during playing in all groups, IOP returned to baseline within 30 seconds after the performance”

Line 375

“Conclusion

No significant differences in IOP were observed before, during or after playing Thai traditional, Thai folk, or Western wind instruments. Although a mild and transient elevation in IOP was noted during performance, IOP levels returned to baseline within 30 seconds after performance was stopped in all groups. These findings suggest that playing Thai and Western wind instruments for a short period of time should not be a concern for increasing the risk of glaucoma in young, healthy individuals. Nevertheless, caution should be warranted when interpreting these results in individuals with glaucoma or glaucoma suspects.”

Explanation

Thank you for your valuable comments. We rearranged the conclusions in the abstract and manuscript to make them more consistence with the primary outcome.

Response from reviewer

2. Although the sample size was calculated to include five participants per group, based on a power analysis, a total of 15 participants is a relatively small sample size. This raises concerns that inter-subject variability may not be adequately captured. It is also possible that the effect size used in the simulation was overestimated, leading to an underestimation of the required sample size. These limitations should be acknowledged in the manuscript to allow readers to appropriately interpret the findings.

New/edit sentence in manuscript

Line 362

“Additionally, the sample size of five participants per group (total n =15) is relatively small. Although this sample size achieved a statistical power of 0.88 to detect differences in IOP among the three groups, caution is warranted because this limited number might not fully capture inter-subject variability. Furthermore, there is a possibility that the effect size from the simulated data that used for power calculation was overestimated, potentially leading to an underestimation of the required sample size”

Explanation

Thank you for your valuable comment. We have incorporated a statement regarding the sample size caution in the limitations section of the manuscript.

Reviewer #6:

Response from reviewer

1. Due to the consideration of gender as a factor affecting IOP changes during performance, the author should be recommended to explain whether baseline data will impact the study results in the Thai folk group, where all performers are male, compared to the other two groups.frequency

New/edit sentence in manuscript

Line 308

“…and sex.”

Line 313

“Notably, the TF group consisted entirely of male participants, whereas the other groups included both genders. While no statistically significant difference in sex distribution was observed among groups (P = 0.08), potential sex-related effects on IOP warrant cautious interpretation [13]. Therefore, sex was included as a covariate in the statistical analysis to adjust for its potential confounding effect.”

Explanation

Thank you for your valuable suggestion about this issue. We acknowledge that gender may act as a confounding factor, hence, we included gender as a covariate in the analysis to adjust for its potential effect. This condiseration has been incorporated into the discussion part.

Response from reviewer

2. The authors should acknowledge that the results being non-significant could be because participants played the low tone frequency and low playing intensity musical notations, and further supplement studies data with higher strength notes

New/edit sentence in manuscript

Line 302

“These low-intensity settings might also explain the nonsignificant differences in IOP elevation between groups. Further studies employing higher tone frequencies and increased playing intensity might help determine whether these factors affect IOP changes across different instrument groups.”

Explanation

Thank you. We have discussed more about the cause of nonsignificant IOP change between groups (which is the primary outcome of the study) and proposed the idea for further studies.

Response from reviewer

3. The authors did not adjust for the effect of CCT on IOP in their analyses; please redo the analyses after Correction value.

New/edit sentence in manuscript

Line 135

“CCT-adjusted IOP values were calculated using the online IOP calculator [7]. These corrected IOP values were subsequently used for the analysis.”

Line 177

“CCT was higher in the Thai traditional group compared to the others, therefore, CCT-adjusted IOP value was calculated. Corrected IOP values of the right and left eyes did not differ significantly between groups, nor between eyes within any group.”

All outcome values in the manuscript

Table 1: adjusted IOP

Table 2,3: all data

Figure 1A, 1B

Explanation

Thank you for your valuable comment. We performed the analysis again using the corrective value of IOP and found that the new result was consistent with our previous findings. We reported the corrective value and the new outcome in Table 1-3 and Fig1A,1B (the spaghetti plot). The corrective value was calculated by IOP-CCT correction formula referenced by https://www.iop-calculator.com/.

We also revised some sentences to ensure clarity in the discussion section.

Line 292

It can be concluded that Thai wind instruments cause a temporary IOP rising comparable to that observed with Western wind instruments, despite differences in playing and breathing techniques.

Line 325

We assume that this rapid return of IOP after the performance may be less efficient in elderly musicians or those with glaucoma. This is likely due to ischemic optic nerve damage associated with aging and glaucoma, which can impair ocular autoregulation [15]. Consequently, these individuals may experience delayed or inadequate IOP normalization following elevated IOP episodes.

---

## [Decision Letter · Decision Letter 2]

10 Sep 2025

PONE-D-25-12979R2Changes in Intraocular Pressure Before, During, and After Playing Thai Traditional, Thai Folk, and Western Wind InstrumentsPLOS ONE

Dear Dr. Thanapaisal,

Thank you for submitting your revised manuscript to PLOS ONE. After careful consideration, we feel that it has merit but does not fully meet PLOS ONE’s publication criteria as it currently stands. Therefore, we invite you to submit a revised version of the manuscript that addresses the points raised during the review process.

(please see section 6).

We look forward to receiving your revised manuscript.

Kind regards,

Jae Hee Kang

Academic Editor

PLOS ONE

Journal Requirements:

Reviewers' comments:

Reviewer's Responses to Questions

**Comments to the Author**

1. If the authors have adequately addressed your comments raised in a previous round of review and you feel that this manuscript is now acceptable for publication, you may indicate that here to bypass the “Comments to the Author” section, enter your conflict of interest statement in the “Confidential to Editor” section, and submit your "Accept" recommendation.

Reviewer #1: (No Response)

Reviewer #5: All comments have been addressed

2. Is the manuscript technically sound, and do the data support the conclusions?

Reviewer #1: Yes

Reviewer #5: Yes

3. Has the statistical analysis been performed appropriately and rigorously? 

Reviewer #1: Yes

Reviewer #5: Yes

4. Have the authors made all data underlying the findings in their manuscript fully available?

Reviewer #1: No

Reviewer #5: Yes

5. Is the manuscript presented in an intelligible fashion and written in standard English?

Reviewer #1: Yes

Reviewer #5: Yes

6. Review Comments to the Author

Reviewer #1: The authors have been very responsive and have provided a thoughtful revision.

Minor edits:

Abstract: line 39 is incomplete; change to "were not significantly different".

Abstract: line 43 needs a period.

Throughout the text, please do not use the term "elderly" as that is considered a negative descriptor; instead substitute with "older".

Reviewer #5: (No Response)

7. PLOS authors have the option to publish the peer review history of their article (what does this mean? ). If published, this will include your full peer review and any attached files.

**Do you want your identity to be public for this peer review?** For information about this choice, including consent withdrawal, please see our Privacy Policy .

Reviewer #1: No

Reviewer #5: No

---

## [Author Response · Author response to Decision Letter 3]

12 Sep 2025

Please find the attached the rebuttal letter. Thank you.

---

## [Editor Report · Decision Letter 3]

15 Sep 2025

Changes in Intraocular Pressure Before, During, and After Playing Thai Traditional, Thai Folk, and Western Wind Instruments

PONE-D-25-12979R3

Dear Dr. Thanapaisal,

We’re pleased to inform you that your manuscript has been judged scientifically suitable for publication and will be formally accepted for publication once it meets all outstanding technical requirements.

Kind regards,

Jae Hee Kang

Academic Editor

PLOS ONE

---

## [Editor Report · Acceptance letter]

PONE-D-25-12979R3

PLOS ONE

Dear Dr. Thanapaisal,

I'm pleased to inform you that your manuscript has been deemed suitable for publication in PLOS ONE. Congratulations! Your manuscript is now being handed over to our production team.

Kind regards,

on behalf of

Dr. Jae Hee Kang

Academic Editor

PLOS ONE